# The Impact of Kefir Consumption on Inflammation, Oxidative Stress Status, and Metabolic-Syndrome-Related Parameters in Animal Models: A Systematic Review and Meta-Analysis

**DOI:** 10.3390/foods14122077

**Published:** 2025-06-12

**Authors:** Zahid Naeem Qaisrani, Wai Phyo Lin, Bo Bo Lay, Khin Yadanar Phyo, Myat Mon San, Nurulhusna Awaeloh, Sasithon Aunsorn, Rinrada Pattanayaiying, Susakul Palakawong Na Ayudthaya, Choosit Hongkulsup, Nirunya Buntin, Sasitorn Chusri

**Affiliations:** 1Biomedical Technology Research Group for Vulnerable Populations and School of Health Science, Mae Fah Luang University, Chiang Rai 57100, Thailand; engr.zbaloch@gmail.com (Z.N.Q.); 6431804208@lamduan.mfu.ac.th (W.P.L.); 6531804115@lamduan.mfu.ac.th (K.Y.P.); 6531804120@lamduan.mfu.ac.th (M.M.S.); 6551811005@lamduan.mfu.ac.th (S.A.); 2Department of Chemical Engineering, Faculty of Engineering & Architecture, Balochistan University of Information Technology, Engineering and Management Sciences (BUITEMS), Quetta 87300, Balochistan, Pakistan; 3School of Information Technology, Mae Fah Luang University, Chiang Rai 57100, Thailand; 6431306113@lamduan.mfu.ac.th; 4Thai Faculty of Allied Health Science, Nakhon Ratchasima College, Nakhon Ratchasima 30000, Thailand; 6451811004@lamduan.mfu.ac.th; 5Department of Food Innovation and Professional Chef, Faculty of Science and Technology, Suan Sunandha Rajabhat University, Bangkok 10300, Thailand; rinrada.pa@ssru.ac.th; 6Thailand Institute of Scientific and Technological Research (TISTR), Pathum Thani 12120, Thailand; susakul@tistr.or.th; 7Department of Food Science and Technology, Faculty of Science and Technology, Suan Sunandha Rajabhat University, Bangkok 10300, Thailand; choosit.ho@ssru.ac.th; 8College of Innovation and Management, Songkhla Rajabhat University, Songkhla 90000, Thailand; nirunya.bu@skru.ac.th

**Keywords:** kefir, metabolic syndrome, probiotics, rodent model, inflammation, oxidative stress

## Abstract

Metabolic syndrome (MetS) is a complex condition defined by central obesity, insulin resistance, dyslipidemia, and systemic inflammation. Kefir, a fermented beverage rich in probiotics and beneficial compounds, has emerged as a functional food that may offer metabolic advantages. Nevertheless, preclinical results have been variable. This systematic review and meta-analysis aimed to assess the influence of kefir and its derived compositions on parameters associated with MetS, inflammation, and oxidative stress in rodent studies. A comprehensive literature search was conducted in PubMed, Scopus, AMED, and LILACS through June 2024. Eligible studies involving kefir interventions in rodent MetS models were included. Data extraction followed PRISMA guidelines, with the risk of bias assessed using the CAMARADES and SYRCLE tools. Meta-analyses were performed with a random effects model. Thirty-eight studies involving 1462 rodents (mice and rats) were analyzed. Kefir significantly reduced body weight gain in both mice (MD = –3.33; 95% CI: –4.89 to –1.77) and rats (MD = –41.53; 95% CI: –54.33 to –28.72). In mice, triglycerides and LDL-C levels decreased significantly; in rats, kefir lowered total cholesterol and triglycerides. Insulin levels were reduced (MD = –0.69; 95% CI: –1.16 to –0.22), suggesting improved insulin sensitivity. Several studies also reported reductions in TNF-α, IL-1β, and IL-6. Despite promising results, the high heterogeneity and methodological variability emphasize the need for standardized preclinical protocols and clinical validation. These findings support the role of kefir as a functional food for metabolic health promotion.

## 1. Introduction

Metabolic syndrome (MetS) consists of interconnected metabolic abnormalities such as central obesity, insulin resistance, dyslipidemia, and hypertension, leading to a heightened risk of type 2 diabetes, cardiovascular diseases (CVD), and some cancers. It significantly contributes to the ongoing epidemics of diabetes and CVD, with expected increases in these conditions [1,2]. Individuals with MetS are at a considerably greater risk for cardiovascular disease (CVD), experiencing a 32% increased likelihood and a 69.5% higher risk of heart failure [3]. Combining impaired glucose tolerance with MetS further elevates cardiovascular mortality risk, as indicated by a hazard ratio of 2.96 compared to those without MetS [4]. In chronic kidney disease patients, MetS is linked to a 26% higher all-cause mortality risk and a 48% increased risk of cardiovascular events [5]. Globally, the prevalence of MetS ranges from 10% to 50%, with a marked increase in South and Southeast Asia over the last three decades due to urbanization and economic development, shifting the disease burden from communicable to non-communicable diseases [6,7,8]. Chronic low-grade inflammation and oxidative stress are crucial in the pathogenesis of MetS [9]. Obesity, particularly visceral adiposity, heightens pro-inflammatory macrophage polarization in adipose tissue. This leads to increased production of pro-inflammatory cytokines, such as tumor necrosis factor-alpha (TNF-α), interleukin-6 (IL-6), and interleukin-1β (IL-1β), alongside a dysregulated secretion of adipokines, which includes elevated levels of leptin and decreased levels of adiponectin. These changes contribute to systemic inflammation, insulin resistance, and metabolic dysregulation in individuals with MetS [10]. Additionally, oxidative stress causes mitochondrial dysfunction and lipid peroxidation, impairing cellular functions and contributing to MetS components. The interaction between inflammation and oxidative stress creates a vicious cycle that exacerbates metabolic dysfunction in MetS [11].

Fermented foods and beverages containing probiotics or synbiotics, including kefir, yogurt, synbiotic yogurt (fortified with prebiotics), kimchi, kombucha, and sauerkraut, have demonstrated effectiveness in alleviating metabolic syndromes by reducing chronic inflammation and oxidative stress [12]. Key components of metabolic syndrome, including glycemic indices and lipid profiles, show improvement with the consumption of probiotics [12,13,14]. Probiotics enhance gut health by altering the microbiota and producing bioactive metabolites, particularly short-chain fatty acids (SCFAs), which further mitigate oxidative stress. Specific strains, such as *Bifidobacterium lactis*, *Enterococcus faecium*, and *Lactobacillus paracasei* reduce inflammatory markers in both healthy and metabolic-syndrome-afflicted models [15,16]. At present, probiotic beverages, including kefir [17], kombucha [18], drinkable yogurt [19], and kvass [20], have garnered significant attention from consumers. These beverages are regarded as alternative interventions that facilitate the stability of the gut microbiome and demonstrate advantageous effects on MetS, reducing systemic inflammation and oxidative stress. Thus, incorporating these functional beverages into the diet might represent a practical approach for managing MetS alongside conventional medical treatments.

Kefir is a probiotic-rich beverage made from the fermentation of milk or plant-based substrates by a symbiotic mix of bacteria and yeasts known as kefir grains. It includes beneficial microorganisms such as Lactobacillus, Leuconostoc, Lactococcus, and Acetobacter, as well as yeasts like Saccharomyces and Kluyveromyces [17,21]. Kefir also contains bioactive metabolites, including peptides, exopolysaccharides, and organic acids, which enhance its antioxidant, anti-inflammatory, antimicrobial, and hypocholesterolemic effects [17,21,22,23]. Preclinical studies conducted in rodent models have shown their potential to improve lipid profiles, reduce body weight gain, enhance glucose tolerance, and suppress inflammatory cytokines such as TNF-α and IL-6 [24,25,26]. These anti-inflammatory effects are thought to involve multiple pathways of action across various cell types, including the modulation of immune cell signaling, inhibition of NF-κB activation, and regulation of the JAK2 signaling pathway, all of which collectively contribute to the reduction of systemic inflammation. A systematic review highlighted the role of kefir in modulating immune responses and reducing inflammation through the alteration of pro-inflammatory cytokines and enhancement of anti-inflammatory responses, alongside changes in intestinal microbiota composition [17,27,28]. However, the evidence across studies remains fragmented due to methodological variations, including differences in kefir type (e.g., dairy vs. non-dairy, live vs. heat-treated), animal species used, duration of intervention, and measurement of outcomes.

Despite the increasing number of experimental studies, a thorough synthesis of findings from animal research and an assessment of the overall effectiveness of kefir and its components in relation to MetS remains scarce. Systematic reviews and meta-analyses of preclinical data play a crucial role in guiding future investigations, improving experimental designs, and contributing to the translational potential for clinical applications. Therefore, this study aimed to systematically evaluate and quantitatively assess the impacts of kefir and its derivatives, which include kefir grains, kefir-fermented products, isolated probiotics, peptides, exopolysaccharides, and other kefir-derived bioactive compounds, on factors associated with metabolic syndrome, including body weight, lipid levels, glucose and insulin concentrations, and inflammatory and oxidative stress markers in rodent models. The findings from this study are expected to offer significant insights into the therapeutic potential of kefir as a functional food for managing metabolic health.

## 2. Methods

The design and reporting of our study adhered to the Preferred Reporting Items for Systematic Reviews and Meta-Analyses (PRISMA) guidelines [29]. This systematic review was registered in the International Platform of Registered Systematic Review and Meta-analysis Protocols (INPLASY^®^) register (INPLASY2024100010; DOI: 10.37766/inplasy2024.10.0010).

### 2.1. Data Sources and Searches

A comprehensive review of studies was conducted using the online databases PubMed, Scopus, AMED, and LILACS, covering publications up to June 2024 with common keywords related to kefir and metabolic syndromes without language restrictions. All search terms can be found in the Appendix A. Furthermore, we conducted a manual examination of the bibliographies of all selected articles to identify additional pertinent papers that were not included in the initial database search.

### 2.2. Study Selection and Eligibility

All abstracts and titles were screened by three authors (ZNQ, WPL, and BBL) to include papers that addressed kefir consumption in rodent studies related to metabolic syndromes. Duplicate articles across all databases were removed. Additionally, studies involving humans, in vitro studies, letters to the editor, dissertations, theses, and papers not relevant to the main issue were excluded from the study. We included studies that met the following inclusion criteria: (1) use of kefir or isolated bacterial strains and metabolites as monotherapy or in combination with other drugs, chemicals, or types of interventions; (2) utilization of a rodent model for metabolic syndromes; (3) a control group receiving alternative treatments or compared with a placebo; (4) outcomes reported concerning metabolic-syndrome-related parameters (lipid profile, glucose levels, blood pressure, blood glucose, and insulin resistance), inflammation markers (interleukin (IL)-6, IL-1, tumor necrosis factor (TNF)), or oxidative stress markers (MDA, ROS, and GSH levels).

### 2.3. Data Collection

Data related to experimental design were extracted, and for each comparison, the mean, median, and percentage reduction of the specified parameters in both the treated and control groups were recorded. General characteristics (year of publication, author, date of publication, and country), as well as characteristics of the experimental model (animal lineage, number of animals, sex, age, and initial weight) and experimental techniques (number of experimental groups, number of animals per group, metabolic syndrome (MetS) induction, presence of a control group, treatment and dosage, and duration of the intervention), were among the parameters of interest in the included studies. The primary outcome measures for further analysis included direct metabolic parameters (e.g., body weight, glucose, insulin, lipid profiles), inflammatory markers, and oxidative stress status, which were based on the availability of data from the included studies. While cortisol levels and parameters related to the hypothalamic–pituitary–adrenal (HPA) axis are recognized as key factors in regulating metabolic syndrome, a comprehensive review of the included preclinical studies showed no findings regarding these hormonal outcomes. Consequently, cortisol and HPA axis parameters were excluded from this current review. It is important to emphasize that a comprehensive understanding of the role of kefir in modulating stress-related metabolic dysfunction is urgently needed.

In instances where the data were not clearly or comprehensively described, the authors were contacted seeking the information; studies from which no response was received after a two-week period were subsequently excluded. Furthermore, a quality assessment of the studies was conducted utilizing the Collaborative Approach to Meta-Analysis and Review of Animal Data from Experimental Studies (CAMARADES) checklist items [30] (Appendix A).

### 2.4. Quality Assessment of the Included Articles

All articles selected for this review underwent an analysis of the risk of bias utilizing SYRCLE’s Risk of Bias (RoB) tool, specifically designed for animal intervention studies [31], which comprises ten components (Appendix A). These components pertain to six distinct types of bias: selection bias, performance bias, detection bias, attrition bias, reporting bias, and other biases. Terminology signifying low, high, or unclear was employed to delineate each domain. Three researchers (WPL, KYP, and MMS) conducted independent assessments of the quality of the included articles.

### 2.5. Statistical Analysis

A meta-analysis of extracted data, specifically concentrating on parameters associated with metabolic syndrome—such as body weight gain; lipid profiles including total cholesterol, HDL cholesterol, and LDL cholesterol; triglycerides; plasma glucose; and insulin levels—was performed utilizing MetaAnalysisOnline.com (https://metaanalysisonline.com; accessed on 10 January 2025) [32]. The mean values, standard deviations (SD), and sample sizes from both the intervention and control groups were directly entered into the tool. When studies presented standard errors (SE), these were converted into SD prior to data input. Continuous outcomes were analyzed employing a random effects model, due to the anticipated methodological and biological heterogeneity across studies. The mean difference (MD) with a 95% confidence interval (CI) was utilized to ascertain effect sizes, taking into account the various measurement scales applied in the studies. The between-study variance was estimated using the method of moments (DerSimonian and Laird), with the heterogeneity among studies assessed through Cochran’s Q-test and quantified using the I^2^ statistic. An I^2^ value exceeding 50% signified substantial heterogeneity. Publication bias was visually examined through funnel plots generated by MetaAnalysisOnline.com.

## 3. Results

### 3.1. Study Inclusion

The article selection process adhered to the PRISMA guidelines [29], and a PRISMA flow diagram (Figure 1) was created to illustrate the number of articles included or excluded at each stage of the protocol. A total of 120 records were located in electronic databases. The following studies were excluded: 27 due to repetitive content, 49 for failing to meet exclusion criteria based on an evaluation of titles and abstracts, and 6 after a thorough review of full-text articles for reasons such as being review articles, clinical studies, or in vitro experiments, or lacking sufficient outcome information. Finally, 38 studies were included in the final analysis of this article.

### 3.2. Characteristics of the Included Studies

A total of 38 studies involving 1462 animals were included: 457 rats and 1005 mice. The details of the experimental studies are provided in the Appendix A. The studies selected from 2006 [33] to 2024 [34] were conducted across 11 distinct countries. Notably, South Korea emerged as the leading nation, accounting for 29% (*n* = 11) of the studies [25,33,34,35,36,37,38,39,40,41,42], followed by Taiwan with 21% (*n* = 8) [24,43,44,45,46,47,48,49]. Nineteen additional studies were conducted in Turkey (*n* = 5; [26,50,51,52,53]), Canada (*n* = 4; [22,54,55,56]), China (*n* = 2; [57,58]), Indonesia (*n* = 2; [59,60]), Brazil (*n* = 2; [61,62]), Malaysia [63], Egypt [64], Argentina [65], and Tunisia [66], as illustrated in Figure 2.

### 3.3. Experimental Models

In studies using animal models, mice were the experimental animals in 67% of cases (*n* = 24) [22,24,25,34,35,36,37,38,39,40,41,42,43,45,46,47,49,54,55,56,60,62,63,65], while rats were used in 33% (*n* = 14) [26,33,44,48,50,51,52,53,57,58,59,61,64,66]. Most of the animals were male, accounting for 82% (*n* = 32), with female rodents only representing 12% (*n* = 5) [22,55,57,58,66]. Only two studies, or 5% (*n* = 2) [54,63], involved both male and female rodents (Figure 3 and Appendix A). The age of the animals utilized in these studies varied from 3 [26,50] to 22 weeks [35]. The age distribution among the rat models indicates that approximately 50% of the studies employed rats younger than 8 weeks, while 35.71% involved rats aged under 8 weeks, and 14.29% did not specify the age of the rats. The majority (79.17%) of studies involving mice utilized animals younger than 8 weeks, whereas a smaller proportion (20.83%) included mice that were 8 weeks or older.

All 38 studies employed dietary strategies to induce metabolic syndrome, primarily using high-fat diets (HFD), which accounted for 36.84% (14 out of 38 studies). In particular, supplying around 60% of calories from fats was the predominant strategy utilized in six studies [34,40,43,46,47,53]. Other approaches included fructose-rich diets (FRD), high-fructose corn syrup solutions (HFCS), hypercholesterolemic diets, and Western diets (WD). The dietary interventions ranged from straightforward high-fat diets that provided 40% to 60% of calories from fats to more complex combinations, such as high-fat high-fructose (HFHF), atherogenic diets, and high-fat high-sucrose (HFHS) diets (Appendix A and Figure 4).

The dietary induction and treatment duration varied significantly in duration across studies, ranging from a brief 6 days [26] to an extensive 15 weeks [50]. However, most studies typically employed induction methods lasting between 6 and 12 weeks, which reflects a standard approach to effectively induce metabolic syndrome (Appendix A and Figure 4). Interestingly, there are only a few studies that combine dietary methods with chemical or genetic approaches. Two studies used a genetic induction model with ApoE −/− mice [45,49], making them susceptible to severe hypercholesterolemia and atherosclerosis, while one study employed chemical induction using streptozotocin (STZ) [63].

### 3.4. Interventions

A comprehensive overview of the experimental designs and kefir treatment protocols utilized in the 38 eligible preclinical studies that evaluate the effects of kefir on metabolic syndrome in rodent models is presented in Table 1. The research explored various types of kefir and its bioactive ingredients. This includes milk-based kefir grains [26,50,62], commercial kefir products [22,51,52,58], lactic acid bacteria derived from kefir [35], isolated probiotics like *Lactobacillus kefiri* and *Lactobacillus mali* APS1 [37,44], kefir peptides [45,48,49], and various formulations of symbiotic and postbiotic kefir [42,59]. Dosages varied considerably across studies. Liquid kefir administration ranged from low volumes of 0.001 mL/g body weight daily [26] to higher doses of 22 mL/kg body weight daily [62]. Probiotic dosages commonly ranged between 10^6^ to 10^10^ CFU/mouse/day [44,63], while powdered forms of kefir and its active components were generally provided in the range of 0.05 mg/g to 400 mg/kg body weight daily [24,48,49]. Dietary incorporation varied from 0.1% to 10% *w/w* [33,40]. The duration of kefir administration across studies was variable, ranging from a minimum of 3 weeks [60,61] to a maximum of 16 weeks [53]. Most studies used treatments of 6 to 12 weeks, indicating a sufficient period to observe metabolic and physiological effects in animal models of metabolic syndrome. These studies mainly utilized control groups such as water [26], saline [38,43], phosphate-buffered saline (PBS) [40,46], microcrystalline cellulose [39,40], milk [25,37,62], or standard chow diet (SCD) [36,55].

### 3.5. Quality Assessment

The methodological quality of the 38 studies was assessed using the 10-item CAMARADES checklist (Appendix A), which evaluates the internal validity of animal research. Most studies showed moderate to high quality. Key aspects such as peer-reviewed publication, random group allocation, and compliance with animal welfare regulations were noted in nearly all studies, indicating strong adherence to experimental standards. However, blinded outcome assessments and sample size calculations were rarely reported, suggesting a lack of rigor and potential observer bias. Less than half disclosed blinded induction of metabolic syndrome or conflicts of interest, indicating transparency issues. Only a few satisfied all 10 criteria, with most scoring between 6 and 8, suggesting good practices but neglect of reproducibility and blinding elements.

The SYRCLE risk of bias tool was used to assess six domains of bias, including selection, performance, detection, attrition, reporting, and other potential sources. Most studies were rated as low risk for random sequence generation, but allocation concealment was often rated high risk or not reported, indicating inadequate control over group assignment processes. Similarly, few studies employed random housing or blinded care of animals, leading to a high or unclear risk for this domain across the dataset. There was consistently high risk due to the lack of random outcome assessment and blinded outcome assessment, which suggests that the results could be influenced by observer expectations. These were often rated as unclear because of insufficient information regarding incomplete data handling and outcome reporting (Appendix A and Figure 5).

### 3.6. Qualitative Synthesis

A comprehensive summary of the outcomes related to metabolic syndrome, inflammation, and oxidative stress parameters influenced by kefir consumption or its active components in rodent models across 38 studies is provided in the Appendix A (Appendix A) and summarized in Figure 6. Kefir administration resulted in a reduction of body weight gain, indicating a beneficial effect of kefir on managing the weight increase commonly associated with metabolic syndrome (e.g., [26,43,54,56]. Additionally, the majority of studies demonstrated improvements in lipid profiles, including total cholesterol (TC), triglycerides (TG), LDL-C, and HDL-C. The consumption of kefir consistently reduced total cholesterol (e.g., [22,34,58]), triglycerides (e.g., [24,26,33]), and LDL cholesterol (e.g., [25,35,37,46]). HDL cholesterol levels were commonly elevated by kefir administration (e.g., [33,39,66]), though some studies reported reductions (e.g., [45,56,58]). Many studies found kefir administration decreased plasma glucose and insulin levels, suggesting a potential role in improving glucose homeostasis and insulin sensitivity (e.g., [26,46,51,52]). Few studies, however, showed mixed or elevated glucose outcomes [56].

Several studies have reported significant anti-inflammatory effects of kefir, particularly in reducing inflammatory cytokines such as TNF-α, IL-1β, and IL-6. A total of 38.58% (n = 12) of the studies reported that dietary interventions such as fructose, HFD, HFHF, and HFCS led to significant increases in TNF-α levels, which were subsequently attenuated by kefir treatment. Five studies (13.16%) focused on the impact of kefir on IL-1β, and the results indicated that kefir consumption greatly reduced elevated levels of IL-1β caused by various diets, including the AD diet, HFD, and HFCS. Similarly, three studies reported a reduction of IL-6, all noting the effectiveness of kefir in this regard. Specifically, Akar et al., (2021) and Chang et al., (2023) [26,45] highlighted the ability of kefir to significantly reduce TNF-α and IL-1β. Kim et al., (2017) reported mixed outcomes, showing increased TNF-α and IL-1β but decreased IL-6 levels [25,37]. Santanna et al., (2017) and Tung et al., (2020) [49,62] also noted decreases in TNF-α and IL-6 levels. It should be noted that only one study [45] explicitly evaluated oxidative stress markers, reporting that kefir intake significantly reduced levels of MDA and oxidized LDL, suggesting potential antioxidant properties of kefir components.

### 3.7. Quantitative Synthesis

The meta-analysis was performed on synthesized data from multiple preclinical studies examining the effects of kefir interventions on metabolic syndrome (MetS)-related parameters, including weight gain, lipid profiles, plasma glucose, and insulin levels in both mouse and rat models. The included studies for meta-analysis contain similar diets and a consistent number of animals per group (i.e., studies with varying numbers of animals in different groups were excluded) (Appendix A).

The meta-analysis for inflammatory and oxidative stress parameters was excluded due to the limited number of comparable studies. Specifically, a predefined threshold was applied, necessitating the inclusion of at least three independent studies reporting on the same outcome for the meta-analysis. This criterion was established to ensure sufficient statistical power and to avoid the instability of effect estimates that may result from extremely small data sets (Appendix A). The results of the meta-analysis are summarized as follows.

A total of 14 studies were analyzed, encompassing a cumulative total of 129 mice across both experimental and control groups. The analysis was conducted utilizing a random effects model in conjunction with the inverse variance method to compare the mean difference (MD). The results indicate a statistically significant difference between the two groups, with the summarized MD reported as −3.33 and a 95% confidence interval ranging from −4.89 to −1.77. The overall effect test reveals significance at *p* < 0.01. Additionally, substantial heterogeneity was identified (*p* < 0.01), implying inconsistent effects in either magnitude or direction. The I^2^ value denotes that 99% percent of the variability among studies is attributed to heterogeneity rather than random chance. Similarly, weight gain in rats across six studies showed that the summarized MD was −41.53 with a 95% confidence interval ranging from −54.33 to −28.72, showing a significant reduction in weight gain in the experimental group compared to controls (*p* < 0.01). These findings emphasize the potential of kefir in managing diet-induced obesity (Figure 7 and Appendix A).

Figure 8 illustrates that the consumption of kefir enhances glycemic control in murine models, as evidenced by a reduction in fasting plasma glucose and insulin levels. A total of eight studies involving 79 mice (both experimental and control groups) were analyzed for plasma glucose; however, no statistically significant difference between the groups was observed (pooled mean difference −29.62, 95% CI: −86.17 to 26.93), with a heterogeneity of 100% (*p* < 0.01). In a subsequent analysis of five studies for insulin level, each encompassing 49 mice, a significant difference was found (pooled MD −0.69, 95% CI: −1.16 to −0.22, *p* < 0.05); nonetheless, substantial heterogeneity continued to exist (*p* < 0.01, I^2^ = 100%) (Appendix A).

The effects of kefir and its derivatives on lipid profiles—including total cholesterol, triglycerides, low-density lipoprotein (LDL), and high-density lipoprotein (HDL)—were evaluated using random effects meta-analyses based on data from studies conducted exclusively in mice (Figure 9 and Appendix A) and rats (Figure 10 and Appendix A). In a mouse model comprising 14 studies with 123 mice, a meta-analysis assessed total cholesterol levels and found no significant difference between kefir-treated and control groups (MD = −37.84, 95% CI: −99.94 to 24.26). High heterogeneity was noted (*p* < 0.01, I^2^ = 100%). For triglycerides, 11 studies with 96 mice per group indicated a significant reduction in levels with kefir consumption (MD = −13.29, 95% CI: −18.39 to −8.19, *p* < 0.05), although high heterogeneity persisted (*p* < 0.01, I^2^ = 97%). The analysis of low-density lipoprotein (LDL) from ten studies involving 85 mice per group revealed a significant reduction in LDL levels (MD = −34.60, 95% CI: −59.26 to −9.93, *p* < 0.05), accompanied by high heterogeneity (*p* < 0.01, I^2^ = 100%). Conversely, no significant difference was found for high-density lipoprotein (HDL) across 11 studies (MD = −0.38, 95% CI: −6.64 to 5.87), with substantial heterogeneity (*p* < 0.01, I^2^ = 98%).

A subgroup meta-analysis was carried out to assess the impact of kefir and its bioactive compounds on serum lipid levels in rat models of metabolic syndrome. Five studies involving 30 rats from both experimental and control groups indicated a statistically significant reduction in total cholesterol levels following kefir treatment. The pooled mean difference (MD) was −7.91, with a 95% confidence interval ranging from −9.87 to −5.95 (*p* < 0.05). However, there was significant heterogeneity (I^2^ = 99%, *p* < 0.01), suggesting considerable variability across the studies. In a separate analysis of five studies with 32 rats per group for triglycerides, kefir supplementation led to a significant reduction in triglyceride levels, with an averaged MD of −13.62 (95% CI: −23.95 to −3.29, *p* < 0.05). Nevertheless, this analysis also revealed high heterogeneity (I^2^ = 100%, *p* < 0.01), indicating variability in the magnitude and directions of the effects. For LDL, four studies involving 24 rats per group showed that kefir treatment did not significantly change LDL levels, resulting in a pooled MD of −0.20 (95% CI: −1.95 to 1.55), with the overall effect test yielding no significance. Yet, significant heterogeneity was noted (I^2^ = 91%, *p* < 0.01), highlighting inconsistencies among the findings. Lastly, five studies with 32 rats per group investigated the effects of kefir on HDL levels, revealing no statistically significant difference between the experimental and control groups (MD = 0.93; 95% CI: −2.55 to 4.42). Similar to other lipid parameters, significant heterogeneity was observed (I^2^ = 98%, *p* < 0.01).

The pooled analyses strongly support the metabolic regulatory potential of kefir and its constituents in rodent models of metabolic syndrome. Significant improvements were observed in body weight management, glycemic control, and lipid profile modulation, particularly in reducing total cholesterol and triglycerides. However, effects on LDL-C and HDL-C levels were inconsistent, highlighting a need for further mechanistic exploration and standardized kefir formulations in future research.

## 4. Discussion

This systematic review and meta-analysis present a comprehensive synthesis of preclinical evidence on the impacts of kefir and its bioactive components on MetS-related outcomes in rodent models. The findings strongly support the metabolic regulatory capabilities of kefir, especially its effectiveness in reducing body weight gain, enhancing lipid parameters, and adjusting glycemic and inflammatory markers. These findings underscore the significance of kefir as a promising functional food with potential applications in the dietary management of metabolic syndrome.

The pooled analyses indicated that kefir interventions significantly diminished weight gain in both mice and rats, thereby suggesting its potential as an anti-obesity agent in the context of diet-induced metabolic dysfunction. This aligns with previous studies indicating that probiotics in fermented foods may reduce adiposity by influencing lipid metabolism and energy balance through interactions involving the microbiota–gut–brain axis, mediated by neural, hormonal, and immune pathways shaped by gut microbiota composition [67,68,69]. Probiotics can enhance gut microbiota and stimulate the production of SCFAs such as acetate, propionate, and butyrate, which is critical for regulating metabolism, glucose, and energy homeostasis [70]. For example, studies indicate that specific probiotic strains, such as *Faecalibacterium prausnitzii*, may aid in weight management and lipid metabolism, especially in high-fat diet models [71].

This review highlights studies that have demonstrated the influence of probiotics and peptides derived from kefir on weight gain, which may involve the modulation of metabolic pathways, such as PPARγ and AMPK. Additionally, kefir, kefir-derived probiotics, and peptides possibly exert these effects by promoting beneficial shifts in gut microbiota and enhancing the production of SCFAs, particularly butyrate [22,26,36,54]. While direct evidence from included animal studies remains limited, previous in vitro studies suggest that SCFAs, which are metabolites enhanced by probiotic fermentation, can activate peroxisome proliferator-activated receptor gamma (PPARγ) [72] and AMP-activated protein kinase (AMPK) pathways [73]. Kefir-derived bioactive peptides have been shown to activate AMPK signaling and upregulate PPARγ expression in cultured hepatocytes and adipocytes, contributing to enhanced lipid oxidation and decreased lipogenesis [24,36,37]. The consumption of kefir peptides reduces body weight and fat accumulation in obesity models induced by high-fat diets. It could possibly enhance lipid metabolism by inhibiting lipogenesis and promoting fatty acid oxidation via increased liver phosphorylated AMPK and PPARα expression [36,48]. Moreover, PPAR agonists that kefir activates can influence AMPK activity, indicating a synergistic effect for metabolic regulation [74]. Therefore, its anti-obesity effects are fundamentally linked to these metabolic pathways.

Furthermore, notable reductions in triglyceride levels [24,34,35,36,37,38,41,42,43,46,63] and LDL cholesterol levels [35,36,37,38,40,41,42,46,49,63] have been observed due to kefir consumption, particularly in mouse models. However, mixed results were found regarding total cholesterol levels. A significant reduction was noted in rats [33,48,57,58,66], while no statistically significant effect was observed in mice [22,24,34,35,36,37,38,40,42,46,49,63]. Moreover, no consistent or significant effects concerning HDL levels were detected across models. Clinical trials have also indicated that kefir consumption can lead to reductions in LDL cholesterol and triglycerides, particularly in individuals with dyslipidemic conditions, although the effects may vary depending on the specific microbial composition of the kefir consumed [75]. The presence of *Lactobacillus plantarum* in kefir plays a significant role in cholesterol reduction by exhibiting bile salt hydrolase (BSH) activity, which hydrolyzes bile salts, leading to the precipitation of cholesterol and its subsequent removal from the body [76,77]. Kluyveromyces strains, in particular *Kluyveromyces marxianus* and *Kluyveromyces lactis,* have demonstrated high cholesterol-reducing capabilities due to their efficient BSH activity, which facilitates the breakdown of bile salts and cholesterol [78].

In murine models, significantly decreased insulin levels were also noted, suggesting improved insulin sensitivity. Some studies reported reductions in glucose levels, but the overall impact was insignificant. Clinical studies indicate that kefir supplementation significantly lowers fasting blood glucose (FBG) and glycated hemoglobin (HbA1c) levels while increasing C-peptide levels, which suggests enhanced insulin secretion and sensitivity [79,80]. In contrast, other research found that kefir can notably reduce fasting insulin levels and insulin resistance (HOMA-IR) without impacting FBS or HbA1c levels [81]. Kefir influences insulin and glucose regulation through the insulin signaling pathway, primarily via its probiotic components, such as lactic acid bacteria and *Bifidobacterium* spp [82]. Kefir has been shown to enhance glucose uptake in insulin-responsive muscle cells by activating the phosphatidylinositol 3-kinase (PI 3-kinase) pathway [83], which is a critical component of the insulin signaling cascade. This activation leads to glucose transporter 4 (GLUT4) translocation to the cell membrane, thereby increasing glucose uptake.

The probiotic bacteria present in kefir not only contribute to its hypolipidemic and hypoglycemic properties but also serve to reduce oxidative stress and inflammatory markers [84,85]. Indeed, a principal strength of this analysis resides in the evidence substantiating the anti-inflammatory properties of kefir. Multiple preclinical studies demonstrate that kefir administration reduces pro-inflammatory cytokine levels, including TNF-α, IL-1β, and IL-6, in rodent models of metabolic syndrome. For example, Akar et al., (2021) [26] and Chang et al., (2023) [45] reported significant decreases in TNF-α and IL-1β levels in high-fructose-fed and atherogenic-diet-fed rats and ApoE knockout mice, respectively, after kefir or kefir peptide supplementation. Kim et al., (2017) [25] and Kim et al., (2021) [37] observed mixed outcomes, with reductions in IL-6 but varying effects on TNF-α and IL-1β in high-fat-diet-induced obese mice treated with kefir-derived probiotics. Similarly, Santanna et al., (2017) [62] and Tung et al., (2020) [49] found significant decreases in TNF-α and IL-6 levels following kefir intervention in LDL-receptor-deficient mice and diet-induced obesity models. These studies involved various kefir formulations, including whole kefir, isolated peptides, and kefir-derived probiotics, administered over treatment periods ranging from 3 to 16 weeks at doses between 0.05 mg/g and 22 mL/kg body weight. This anti-inflammatory profile aligns with proposed mechanisms in which kefir-derived metabolites interact with toll-like receptors and nuclear factor-kappa B (NF-κB) signaling pathways to suppress inflammatory responses [86]. However, despite the presence of promising findings, the quantitative synthesis of inflammatory and oxidative stress parameters remains constrained due to the limited number of studies reporting comparable outcomes. Kefir exhibits radical-scavenging effects, chelates ferrous ions, and boosts the activity of antioxidant enzymes such as superoxide dismutase (SOD) and catalase. These functions play a crucial role in neutralizing reactive oxygen species (ROS), thereby enhancing their protective properties against oxidative stress [28,63,84,87,88].

Kefir consumption demonstrates considerable anti-inflammatory effects, especially in regulating the immune response [17,28,49]. It decreases pro-inflammatory cytokines while elevating anti-inflammatory cytokines, thus achieving a balance between Th1 and Th2 responses. It additionally promotes the abundance of beneficial bacteria, including Lachnospiraceae and Roseburia, which are recognized for their capacity to generate SCFAs with anti-inflammatory properties [28,82]. These SCFAs regulate intestinal pH, improve barrier integrity, and affect immune responses by activating signaling pathways that are involved in the synthesis of host defense peptides [89]. The polysaccharide extract from kefir, known as kefiran, was found to possess anti-inflammatory properties by inhibiting granuloma formation and reducing paw edema in animal models [90]. Kefir peptides, Kef-1, have demonstrated an ability to inhibit pathways such as NF-κB and MAPK, which play a role in inflammation [91].

The heterogeneity (I^2^) value observed across the meta-analyses of preclinical studies was consistently high, ranging from 91% to 100%. Differences in study design, animal species, sex, age, induction methods for MetS, kefir strain composition, dosage, and treatment duration all contributed to the variability observed in the outcomes. Furthermore, the quality assessment revealed that several studies were deficient in adequate blinding, randomization, and transparency regarding sample size calculations, which may have introduced bias and compromised the reliability of the results. Additionally, the elevated heterogeneity observed in the present meta-analysis may be partially influenced by publication bias, particularly the inclination for studies yielding positive or statistically significant results to be more frequently published.

Another major methodological limitation identified in this review was the significant inconsistency in how outcomes were reported in preclinical studies. Firstly, various outcomes related to inflammatory and oxidative stress markers could not be quantitatively analyzed due to an insufficient number of eligible studies (i.e., less than three studies reporting the same outcome). Secondly, an imbalance in sex and age was noted among the included studies. The majority of experiments were conducted on male rodents (82%), with relatively few studies involving female animals. Many studies exhibited considerable variation in the age of animals at the beginning of the study. Sex-specific hormonal and metabolic differences can significantly affect responses to interventions for metabolic syndrome, including changes in lipid metabolism, insulin sensitivity, and inflammatory pathways. Similarly, the stage of development or aging may influence metabolic outcomes and the effectiveness of kefir interventions. Consequently, these factors may limit the generalizability of the findings across both sexes and various life stages. Future preclinical studies should include balanced sex representation and standardized age groups to enhance the translational relevance of the results for broader populations.

Despite these limitations, the conclusions drawn from this investigation carry significant implications. Primarily, they advocate for incorporating kefir and its derived components into functional food frameworks for managing MetS. Furthermore, the findings underscore the necessity for enhanced standardization in the production of kefir products, particularly regarding microbial content and bioactive metabolites. This standardization is crucial for ensuring both reproducibility and efficacy. Lastly, the results provide a scientific basis for transitioning these findings into clinical trials involving human participants. Conducting randomized controlled trials among individuals diagnosed with MetS or its related components is essential to authenticate the preclinical findings regarding the efficacy of kefir and to elucidate its safety and long-term health implications.

## 5. Conclusions

In conclusion, this systematic review and meta-analysis provide robust preclinical evidence supporting the beneficial effects of kefir on weight management, lipid modulation, and insulin regulation, as well as the reduction of oxidative stress and inflammation in rodent models of metabolic syndrome. Although the findings are promising, notable heterogeneity and methodological limitations prevalent across the studies highlight the necessity of more standardized investigations. Furthermore, due to the complex composition of kefir, which contains multiple strains of probiotics, bioactive peptides, exopolysaccharides, and organic acids, additional mechanistic studies are warranted to clarify the specific pathways and molecular targets responsible for its beneficial effects. Kefir continues to be a compelling candidate within the functional food domain for the prevention and management of metabolic syndrome, thereby warranting further exploration in both translational and clinical contexts.

## Figures and Tables

**Figure 1 foods-14-02077-f001:**
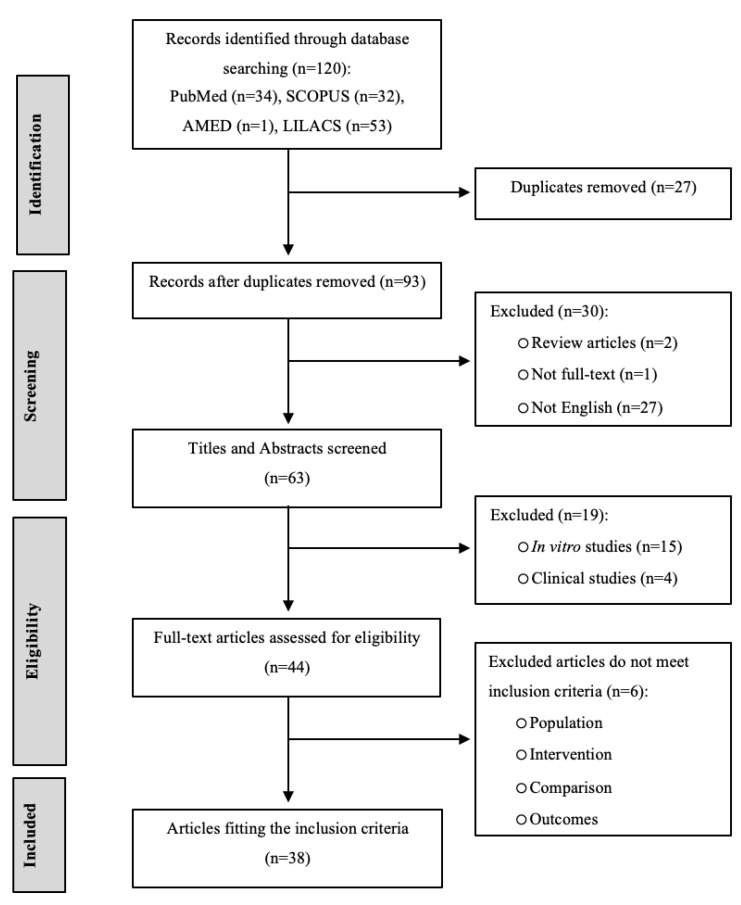
PRISMA flow diagram of the systematic review process.

**Figure 2 foods-14-02077-f002:**
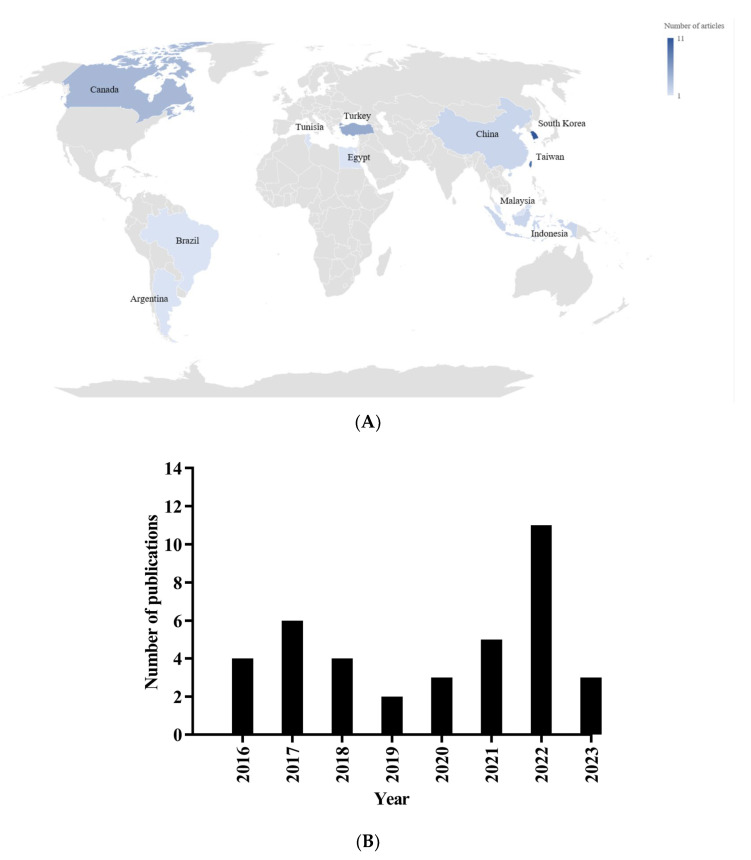
Geographical distribution (**A**) and publication year (**B**) of included studies.

**Figure 3 foods-14-02077-f003:**
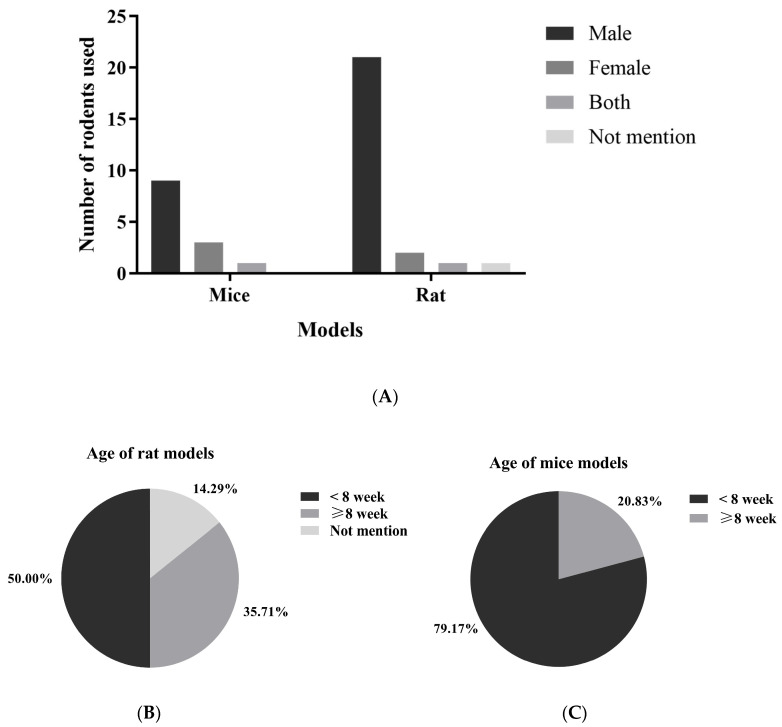
The gender (**A**) and age (**B**,**C**) distribution of rodents used in kefir evaluation studies represent the demographics of rodents used to assess the beneficial effects of kefir in studies involving metabolic syndrome.

**Figure 4 foods-14-02077-f004:**
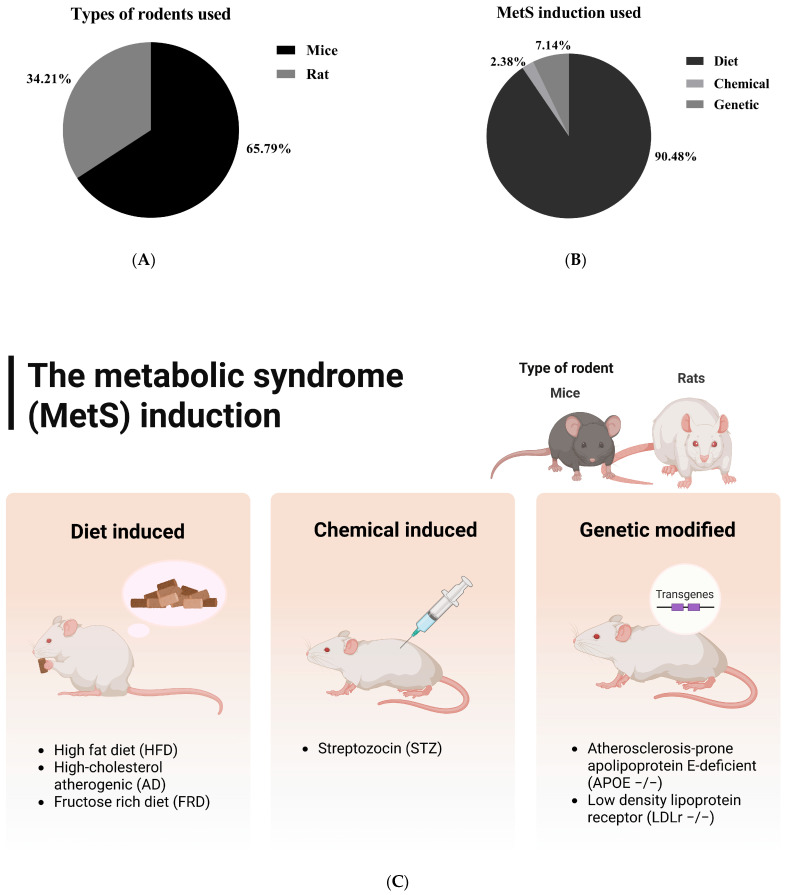
Types of rodents used in the included studies (**A**) and the detailed methods (**B**,**C**) for inducing metabolic syndrome in these rodents. The first group represents diet-induced metabolic syndrome, while the second and third models combine dietary and chemical induction (e.g., high-fat diet plus streptozotocin) and dietary and genetic induction (e.g., ApoE knockout mice), respectively. This classification reflects the experimental strategies utilized in the included studies.

**Figure 5 foods-14-02077-f005:**
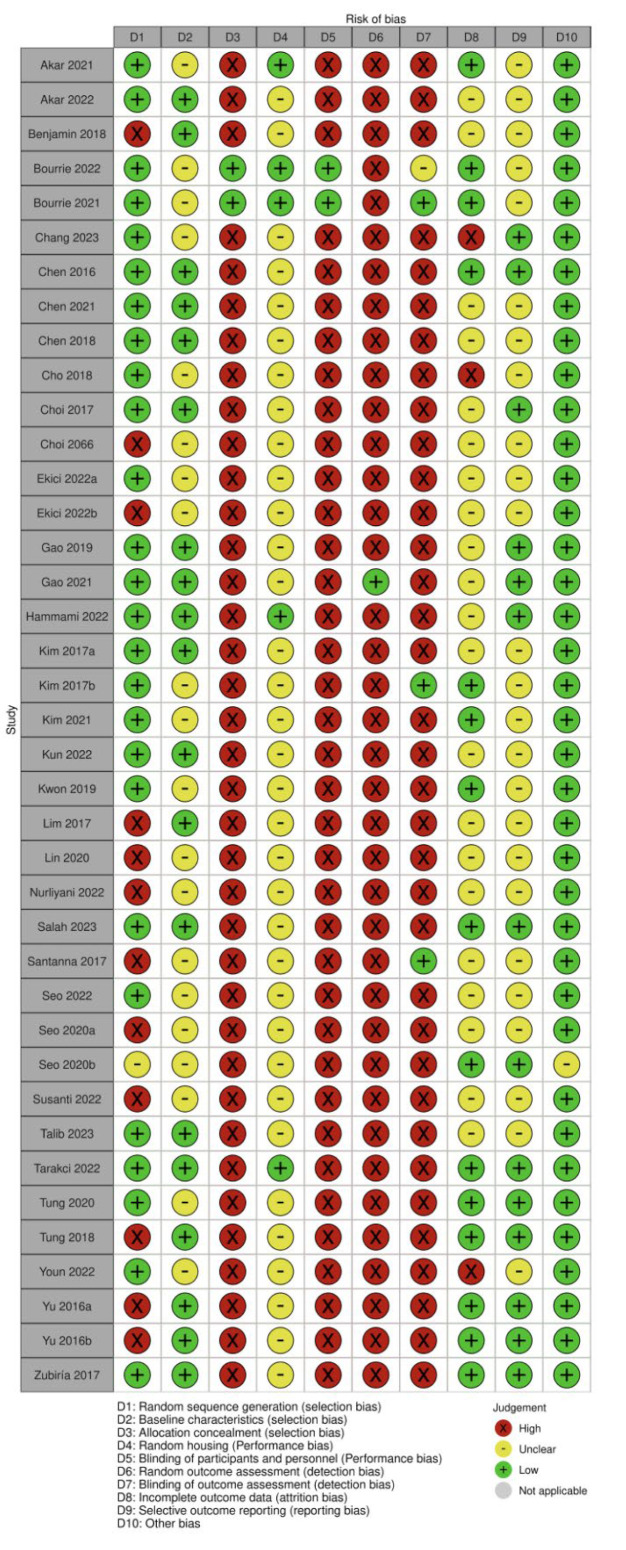
Quality of reporting and bias evaluation conducted with SYRCLE’s risk of bias tool. The upper panel illustrates the quality of reporting and bias risk in the included studies [22,24,25,26,33,34,35,36,37,38,39,40,41,42,43,44,45,46,47,48,49,50,51,52,53,54,55,56,57,58,59,60,61,62,63,64,65,66], while the lower panel evaluates biases related to selection, performance, detection, attrition, and other factors.

**Figure 6 foods-14-02077-f006:**
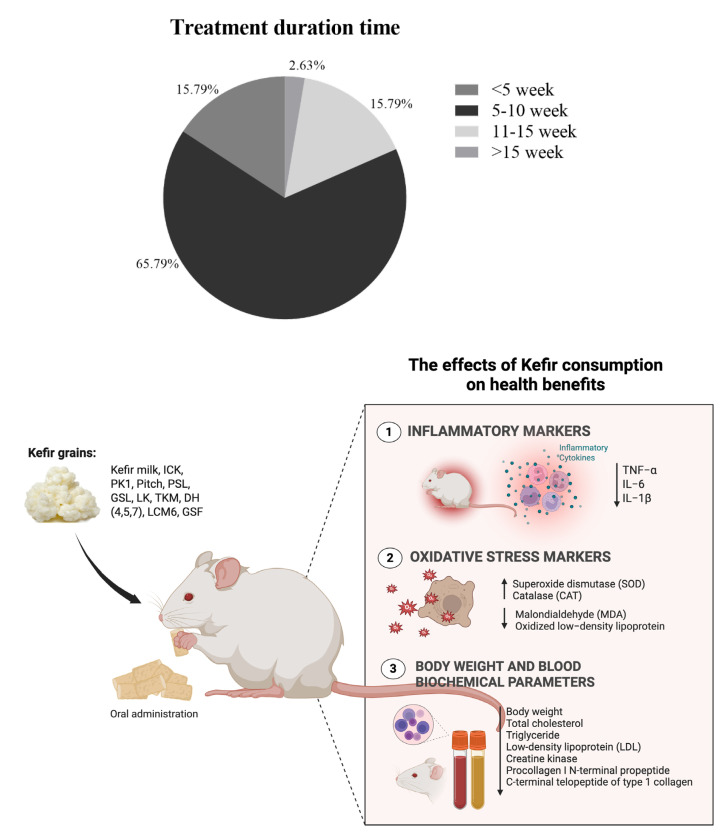
Overview of study characteristics and outcome measures. The upper panel summarizes the distribution of treatment durations across the included studies, while the lower panel presents the inflammatory and oxidative stress markers evaluated in those studies.

**Figure 7 foods-14-02077-f007:**
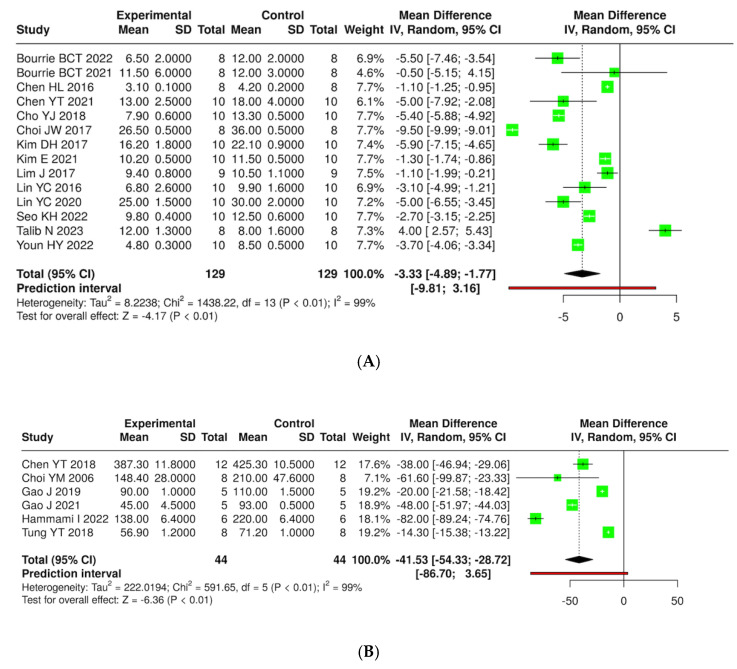
Forest plot analysis presenting the effects of consuming different types of kefirs, their isolated bacteria, or active components, compared to the control group, on weight gain in mouse (**A**) [24,25,35,36,38,40,41,42,43,46,47,54,55,63] and rat (**B**) [36,44,48,57,58,66] models. Each green square shows the mean difference (MD) point estimate for individual studies, while the red line denotes the 95% prediction interval for the overall meta-analysis.

**Figure 8 foods-14-02077-f008:**
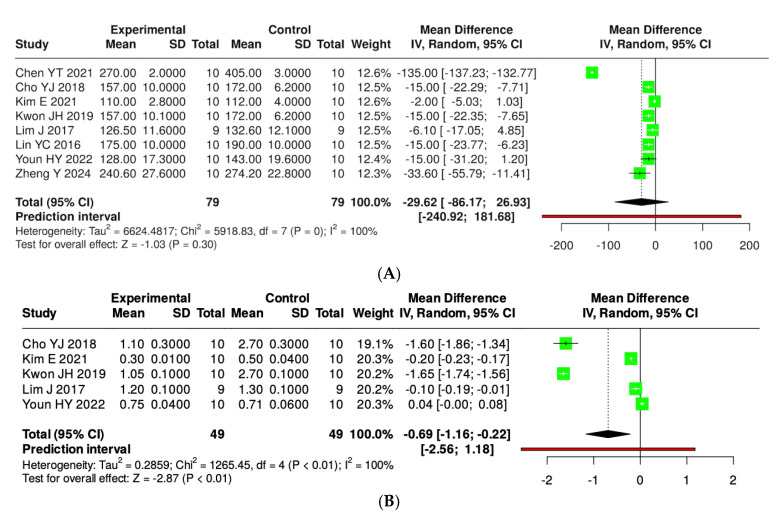
Forest plot analysis showing the effects of consuming various types of kefirs, their isolated bacteria, or active components, compared to the control group, on plasma glucose (**A**) [34,35,38,39,40,42,43] and insulin (**B**) [35,38,39,40,42] levels in mouse models. Each green square shows the mean difference (MD) point estimate for individual studies, while the red line denotes the 95% prediction interval for the overall meta-analysis.

**Figure 9 foods-14-02077-f009:**
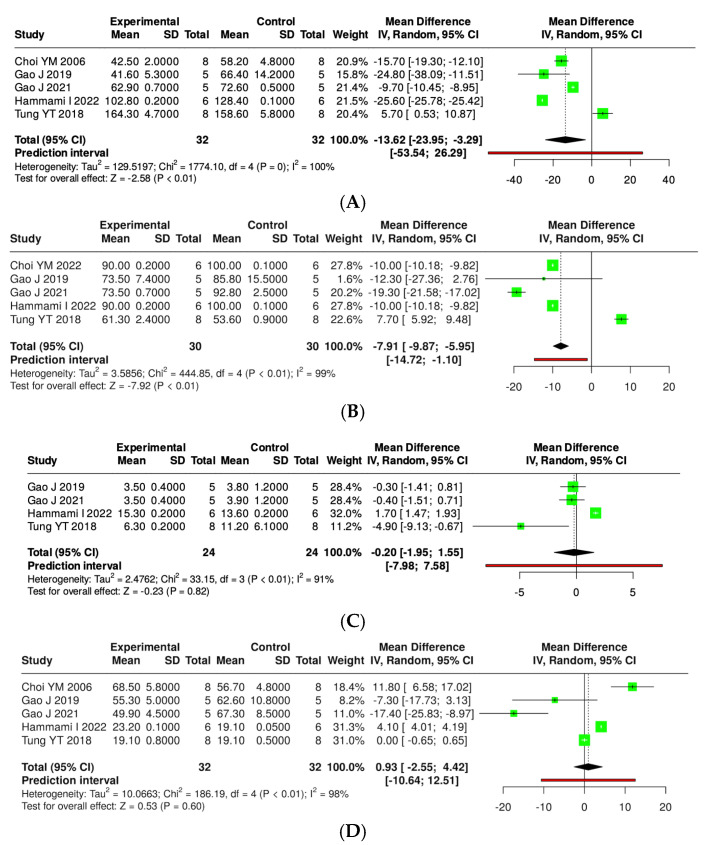
Forest plot analysis illustrating the effects of consuming various types of kefirs, their isolated bacteria, or active components, in comparison to the control group, on lipid profiles, including levels of total cholesterol (**A**) [36,48,57,58,66], triglycerides (**B**) [36,48,57,58,66], low-density lipoprotein (**C**) [48,57,58,66], and high-density lipoprotein (**D**) [36,48,57,58,66] in rat models. Each green square shows the mean difference (MD) point estimate for individual studies, while the red line denotes the 95% prediction interval for the overall meta-analysis.

**Figure 10 foods-14-02077-f010:**
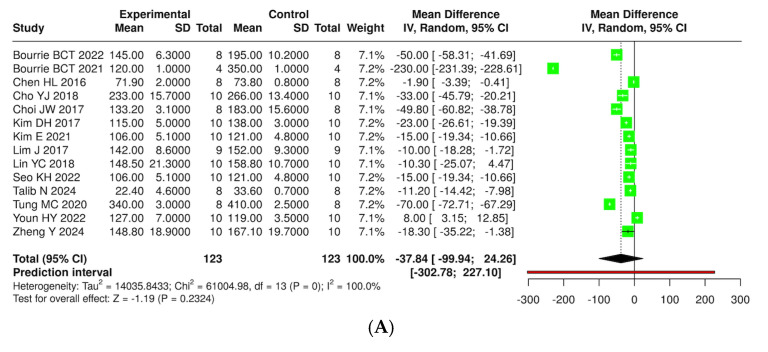
Forest plot analysis illustrating the effects of consuming various types of kefirs, their isolated bacteria, or active components, in comparison to the control group, on lipid profiles, including levels of total cholesterol (**A**) [24,25,34,35,36,38,40,41,42,46,49,54,55,63], triglycerides (**B**) [24,25,34,35,36,38,41,42,43,46,63], low-density lipoprotein (**C**) [25,35,36,38,40,41,42,46,49,63], and high-density lipoprotein (**D**) [25,35,36,38,40,42,46,49,54,55,63] in mouse models. Each green square shows the mean difference (MD) point estimate for individual studies, while the red line denotes the 95% prediction interval for the overall meta-analysis.

**Table 1 foods-14-02077-t001:** Experimental designs and treatment protocols for animal models of metabolic syndromes used in the eligibility studies.

Author, Year [Reference]	Types of Intervention *	Dose	Control Group ^#^	Testing Duration (Weeks)
1. Akar et al., 2021 [26]	Milk-based kefir grains	0.001 mL/g bw, daily	Water	6
2. Akar et al., 2022 [50]	Milk-based kefir grains	0.01 mL/g bw, daily	Water	6
3. Angelis-Pereira et al., 2013 [61]	Fermented kefir (kefir grains with distilled water and 5% brown sugar)	8.6 mg/g bw, daily	Water	3
4. Bourrie et al., 2018 [22]	Commercial kefir (grain)	100 mL, daily	Milk	12
5. Bourrie et al., 2022 [54]	Pitched kefir (fermented kefir grains and 2% fat milk with a mixture of microbes)	3.1 g/mouse, daily	LFD with milk	8
6. Bourrie et al., 2021 [55]	Pitched kefir (fermented kefir grains and 2% fat milk with a mixture of microbes)	2 mL kefir/20 g food, daily	SCD	8
7. Chang et al., 2023 [45]	Kefir peptides powder (KEFPEP^®^) containing 23.1 g of peptides per 100 g powder	0.328 mg/g bw (for low dose) 0.655 mg/g bw (for high dose)	SCD	13
8. Chen et al., 2016 [24]	Kefir peptides powder	0.05, 0.10, 0.15 mg/g bw	Water	8
9. Chen et al., 2021 [43]	AB-kefir (commercial)	10^9^ CFU/mouse//day	Saline	10
10. Chen et al., 2018 [44]	*Lactobacillus mali* APS1	5 × 10^7^, 5 × 10^8^, and 5 × 10^9^ CFU/mouse//day (for low, medium, and high doses)	Saline	12
11. Cho et al., 2018 [35]	Kefir-derived lactic acid bacteria	10 mL/kg bw	Saline	5
12. Choi et al., 2017 [36]	Commercial kefir powder	0.1% (*w*/*w*) kefir powder in HFD 0.2% (*w/w*) kefir powder in HFD	SCD	8
13. Choi et al., 2006 [33]	Lactic-F (The ferment of kefir gain)	10% (*w/w*) of the ferment in HFD	SCD	4
14. Ekici et al., 2022 [51]	Commercial kefir	10 mL/kg bw	SCD	8
15. Ekici et al., 2022 [52]	Commercial kefir	10 mL/kg bw	Saline	8
16. Gao et al., 2019 [58]	Tibet kefir milk (TKM)	18 mL/kg bw	Water	8
17. Gao et al., 2021 [57]	Tibet kefir milk (TKM)	18 mL/kg bw	Water	4 (TKM2) and 8 (TKM1)
18. Hammami et al., 2022 [66]	Kefir milk	10 mL/kg bw	Semi-skimmed cow milk	8 (+4 days)
19. Kim et al., 2017 [37]	Kefir isolates (*Leuconostoc mesenteroides* (DH4) and *Lactobacillus kefiri* (DH5 and DH7))	0.2 mL of 2 × 10^8^ CFU/mouse//day	Saline	6
20. Kim et al., 2017 [25]	Kefir milk	0.2 mL/mouse//day	Sterilized milk	12
21. Kim et al., 2021 [38]	Cell surface layer proteins from the kefir probiotic lactic acid bacteria	120 mg/kg bw	Saline	6
22. Kwon et al., 2019 [39]	*Leuconostoc mesenteroides* and *Lactobacillus kefiri* isolated from kefir fermented milk	120 mg/kg bw	Microcrystalline cellulose	5
23. Lim et al., 2017 [40]	Water-soluble exopolysaccharides (EPS) from the probiotic kefir and kefir-gain residue (Res)	5% (*w/w*) EPS in HFD 8% (*w/w*) Res in HFD	Microcrystalline cellulose	4
24. Lin et al., 2016 [46]	*Lactobacillus kefiranofaciens* M1 *and Lactobacillus mali* APS1	1 × 10^8^ CFU/mouse/day	PBS	8
25. Lin et al., 2020 [47]	*Lactobacillus kefiranofaciens* M1 *and Lactobacillus mali* APS2	1 × 10^8^ CFU/mouse/day	PBS	8
26. Nurliyani et al., 2022 [59]	Symbiotic kefir and probiotic kefir	18 mL/kg bw/day	NM	4
27. Salah et al., 2023 [64]	Ready-made milk kefir, commercial	1.8 mL/rat/day	NM	12 (early probiotic treated) 3 (late probiotic treated)
28. Santanna et al., 2017 [62]	Milk-based kefir grains	22 mL/kg bw	Soluble fraction of milk	4
29. Seo et al., 2022 [41]	Surface layer protein (SLP) and exopolysaccharides (EPS) from the probiotic kefir	125 mg/kg bw (SLP) 250 mg/kg bw (EPS)	Saline	6
30. Seo et al., 2020 [56]	Heat-killed lactic acid bacteria isolated from kefir grain	10 mL/kg bw, daily	Microcrystalline cellulose	8
31. Susanti et al., 2022 [60]	Goat’s milk kefir	0.52 mL/mouse/day	water	3
32. Talib et al., 2024 [63]	*Lacticaseibacillus paracasei* Isolated from *Malaysian water kefir grains*	1 × 10^6^ CFU/mL/day (a low dose) 1 × 1010 CFU/mL/day (a high dose)	SCD	4
33. Tarakci et al., 2022 [53]	Commercial kefir	6 mL/kg bw (3 days a week)	NM	16
34. Tung et al., 2020 [49]	Kefir peptides powder containing 23.1 g of peptides per 100 g powder	100 mg/kg bw (for low dose) 400 mg/kg bw (for high dose)	PBS	12
35. Tung et al., 2018 [48]	Kefir peptides powder containing 23.1 g of peptides per 100 g powder	164 mg/kg bw (for low dose)	Milk powder	8
36. Youn et al., 2022 [42]	*Lentilactobacillus kefiri* DH5 Bioconversion media (postbiotics)	1 × 10^8^ CFU/kg bw/day 10 mL/kg bw/day	Saline	5
37. Zheng et al., 2024 [34]	Manufactured fermented food kefir *Lactiplantibacillus plantarum* TWK10	5 × 10^8^ CFU/day (a low dose) 1 × 10^9^ CFU/day (a medium dose) 5 × 10^9^ CFU/day (a high dose)	PBS	8
38. Zubiría et al., 2017 [65]	*Lactobacillus kefiri* isolated from *kefir fermented milk CIDCA 8348*	1 × 10^8^ CFU/mouse/day	Milk	6

* Types of intervention specify whether kefir was administered as a whole fermented product, isolated bacterial strains, isolated kefir-derived compounds, or postbiotic preparation. ^#^ LFD: low-fat diet, HFD: high-fat diet, PBS: phosphate buffer saline, SCD: standard chow diet, NM: not mentioned.

## Data Availability

The raw data supporting the conclusions of this article will be made available by the authors on request.

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
