# Peer review of "The Impact of Kefir Consumption on Inflammation, Oxidative Stress Status, and Metabolic-Syndrome-Related Parameters in Animal Models: A Systematic Review and Meta-Analysis"

_foods, 2025, doi:10.3390/foods14122077_

Round 1
Reviewer 1 Report
Comments and Suggestions for Authors
The current paper was a systematic review based on meta-analysis. The reivew provide important information about the impact of kefir consumption on inflammation, oxidative stress status, and metabolic syndrome-related parameters in animal models. Overall, the study was well-designed and written in good english. The paper fall in the scope of Foods and can be considered for publication after revisions.
- In figure 2, Taiwan should be changed as Chinese Taipei.
- In the conclusion part, the authors should mention that further study is also needed to clarify the mechanism of the benefical effects of kefir consumpiton due to the complex components.
- In figure 3, for animal trial, the age of animal should be mentioned. if not, the academic value of the references is low.
- In the discussion, apart from SCFA, other metabolits derived from kefir consumption should be stated.
Author Response
The current paper was a systematic review based on meta-analysis. The reivew provide important information about the impact of kefir consumption on inflammation, oxidative stress status, and metabolic syndrome-related parameters in animal models. Overall, the study was well-designed and written in good english. The paper fall in the scope of Foods and can be considered for publication after revisions.
Reviewer #1-1 In figure 2, Taiwan should be changed as Chinese Taipei.
Response: Taiwan, not Chinese Taipei, was recorded as the Country of publication (see General Characteristics of Studies)
Reviewer #1-2 In the conclusion part, the authors should mention that further study is also needed to clarify the mechanism of the beneficial effects of kefir consumption due to the complex components.
Response: The conclusion has been revised as recommended.
“In conclusion, this systematic review and meta-analysis provide robust preclinical evidence supporting the beneficial effects of kefir on weight management, lipid modulation, insulin regulation, as well as the reduction of oxidative stress and inflammation in rodent models of metabolic syndrome. Although the findings are promising, notable heterogeneity and methodological limitations prevalent across the studies highlight the necessity for more standardized investigations. Furthermore, due to the complex composition of kefir, which contains multiple strains of probiotics, bioactive peptides, exopolysaccharides, and organic acids, additional mechanistic studies are warranted to clarify the specific pathways and molecular targets responsible for its beneficial effects. Kefir continues to be a compelling candidate within the functional food domain for the prevention and management of metabolic syndrome, thereby warranting further exploration in both translational and clinical contexts.”
Reviewer #1-3, In figure 3, for animal trials, the age of the animal should be mentioned. if not, the academic value of the references is low.
Response: The age of the animals used in the included studies is mentioned in Figure 3.
Reviewer #1-4, In the discussion, apart from SCFA, other metabolites derived from kefir consumption should be stated.
Response: Other metabolites derived from kefir consumption have been stated as follows.
“This review highlights studies that have demonstrated the influence of probiotics and peptides derived from kefir on weight gain, which may involve the modulation of metabolic pathways, such as PPARγ and AMPK. Additionally, kefir, kefir-derived probiotics, and peptides possibly exert these effects by promoting beneficial shifts in gut microbiota and enhancing the production of SCFAs, particularly butyrate [22,26,36,54]. While direct evidence from included animal studies remains limited, previous in vitro studies suggest that SCFAs, which are metabolites enhanced by probiotic fermentation, can activate peroxisome proliferator-activated receptor gamma (PPARγ) [72] and AMP-activated protein kinase (AMPK) pathways [73]. Kefir-derived bioactive peptides have been shown to activate AMPK signaling and upregulate PPARγ expression in cultured hepatocytes and adipocytes, contributing to enhanced lipid oxidation and decreased lipogenesis [24, 36, 37].”
Reviewer 2 Report
Comments and Suggestions for Authors
Please find my suggestions and comments in the attached PDF document.

Author Response
This manuscript presents a comprehensive and methodologically sound systematic review and meta-analysis on the effects of kefir on metabolic syndrome-related parameters in rodent models. The review is well-organized, follows PRISMA guidelines, and provides valuable insights into the preclinical evidence supporting kefir's metabolic benefits. Τhere is as a clear adherence to PRISMA guidelines, Risk of bias assessment using both CAMARADES and SYRCLE tools and a thorough review of relevant parameters including body weight, lipid profile, glucose, insulin, and inflammatory markers. Some general suggestions for improvement of the text:
- Justification for Exclusion of Inflammatory and Oxidative Stress Meta-Analyses: While it is mentioned that these were excluded due to a limited number of studies, it would be beneficial to explicitly state the minimum threshold used (e.g., at least three studies). Emphasizing the need for standardized outcome reporting in future studies would strengthen the discussion.
Respond to the reviewer’s comment: Information has been added to the results section, 3.7. Quantitative synthesis and the discussion section are as follows.
The result section:
“The meta-analysis for inflammatory and oxidative stress parameters was excluded due to the limited number of comparable studies. Specifically, a predefined threshold was applied, necessitating the inclusion of at least three independent studies reporting on the same outcome for the meta-analysis. This criterion was established to ensure sufficient statistical power and to avoid the instability of effect estimates that may result from extremely small data sets (Supplementary Table S8).”
The discussion section:
“Another major methodological limitation identified in this review was the significant inconsistency in how outcomes were reported in preclinical studies. Firstly, various outcomes related to inflammatory and oxidative stress markers could not be quantitatively analyzed due to an insufficient number of eligible studies (i.e., less than three studies reporting the same outcome).”
- Sex and Age Imbalance in Animal Models: A clear majority of the included studies used male rodents (82%), and age distribution varied widely. A brief comment on how this sex bias and age variation may affect the generalizability of findings would be welcome.
Respond to the reviewer’s comment: Information has been added to the discussion section as follows.
The discussion section:
“Secondly, an imbalance in sex and age was noted among the included studies. The majority of experiments were conducted on male rodents (82%), with relatively few studies involving female animals. Many studies exhibited considerable variation in the age of animals at the beginning of the study. Sex-specific hormonal and metabolic differences can significantly affect responses to interventions for metabolic syndrome, including changes in lipid metabolism, insulin sensitivity, and inflammatory pathways. Similarly, the stage of development or aging may influence metabolic outcomes and the effectiveness of kefir interventions. Consequently, these factors may limit the generalizability of the findings across both sexes and various life stages. Future preclinical studies should include balanced sex representation and standardized age groups to enhance the translational relevance of the results for broader populations.”
- Mechanistic Insight: The discussion includes important metabolic pathways (AMPK, PPARγ), but could benefit from more focused linkage to the results of included studies. Consider briefly referencing in vitro or mechanistic studies where available.
Respond to the reviewer’s comment: In vitro studies focusing on the cell lines, including hepatocytes, adipocytes, and muscle cell lines, were added to the discussion section.
“Kefir-derived bioactive peptides have been shown to activate AMPK signaling and upregulate PPARγ expression in cultured hepatocytes and adipocytes, contributing to enhanced lipid oxidation and decreased lipogenesis [24, 36, 37]. Additionally, kefir metabolites can modulate insulin signaling by promoting GLUT4 translocation through the activation of the PI3K/Akt pathway in muscle cell lines [83], which complements the observed improvements in glucose and insulin profiles in animal models.”
Some more specific comments:
- Introduction
1.1. In the first paragraph, the sentence “Chronic low-grade inflammation … leading to the production of pro-inflammatory cytokines and adipokines” could be further developed for clarity and depth. I recommend briefly specifying the key adipokines and cytokines involved. For instance, as noted by Savulescu-Fiedler et al., leptin and adiponectin are among the most prominent adipokines, with elevated leptin and reduced adiponectin levels observed in obese individuals. Additionally, the inclusion of representative pro-inflammatory cytokines would strengthen the biological context and support the mechanistic link between low-grade inflammation and metabolic dysregulation.
Respond to the reviewer’s comment: The first paragraph of the introduction has been rewritten as suggested. The text below has been added to the first paragraph.
“Obesity, particularly visceral adiposity, heightens pro-inflammatory macrophage polarization in adipose tissue. This leads to increased production of pro-inflammatory cytokines, such as tumor necrosis factor-alpha (TNF-α), interleukin-6 (IL-6), and interleukin-1β (IL-1β), alongside a dysregulated secretion of adipokines, which includes elevated levels of leptin and decreased levels of adiponectin. These changes contribute to systemic inflammation, insulin resistance, and metabolic dysregulation in individuals with MetS [10].”
1.2. In the second paragraph, the phrase “Probiotic foods, such as synbiotic yogurt and kimchi…” may lead to some confusion, as synbiotic yogurt is more accurately categorized as a synbiotic, not simply a probiotic food. Furthermore, kimchi is only one example among many probiotic-rich fermented foods. I suggest either clarifying the distinction between probiotic and synbiotic foods, or generalizing the category and including additional representative examples (e.g., kefir, kombucha, sauerkraut) to ensure broader coverage and accuracy.
Respond to the reviewer’s comment: The second paragraph of the introduction has been revised as suggested to enhance the clarity. The text below has been incorporated into the second paragraph.
“Fermented foods and beverages containing probiotics or synbiotics, including kefir, yogurt, synbiotic yogurt (fortified with prebiotics), kimchi, kombucha, and sauerkraut, have demonstrated effectiveness in alleviating metabolic syndromes by reducing chronic inflammation and oxidative stress [12].”
1.3. In the third paragraph, the sentence “Preclinical studies conducted in rodent models have demonstrated their potential effect in … suppressing inflammatory cytokines such as TNF-α and IL-6 [24-26]” could be expanded to provide more mechanistic insight. For example, it would be beneficial to mention that these studies suggest multiple pathways of action across different cell types, including modulation of immune cell signaling, inhibition of NF-κB activation and modulation of inflammation via the JAK2 signaling pathway. Such elaboration would enrich the understanding of how these interventions mitigate inflammation at the molecular level.
Respond to the reviewer’s comment: The mentioned information in the third paragraph of the introduction has been revised as suggested to enhance clarity. The text below has been incorporated into the second paragraph.
“Preclinical studies conducted in rodent models have shown their potential to improve lipid profiles, reduce body weight gain, enhance glucose tolerance, and suppress inflammatory cytokines such as TNF-α and IL-6 [24-26]. These anti-inflammatory effects are thought to involve multiple pathways of action across various cell types, including the modulation of immune cell signaling, inhibition of NF-κB activation, and regulation of the JAK2 signaling pathway, all of which collectively contribute to the reduction of systemic inflammation.”
1.4. In the last paragraph of the Introduction, the sentence “Therefore, this study aimed to systematically evaluate and quantitatively assess the impacts of kefir and its derivatives…,” the term “derivatives of kefir” is mentioned but not clearly defined or exemplified anywhere in the manuscript. For clarity and transparency, it would be helpful to specify what these derivatives include—whether they are kefir grains, kefir fermented products, isolated bioactive compounds, or others. I think that this will improve the reader’s understanding of the scope of the study.
Respond to the reviewer’s comment: The sentences were rephrased to clarify the term “derivatives of kefir.”
Therefore, this study aimed to systematically evaluate and quantitatively assess the impacts of kefir and its derivatives, which include kefir grains, kefir-fermented products, isolated probiotics, peptides, exopolysaccharides, and other kefir-derived bioactive compounds on factors associated with metabolic syndrome, including body weight, lipid levels, glucose and insulin concentrations, and inflammatory and oxidative stress markers in rodent models.
- Methods
2.1. The manuscript provides a thorough description of sources, selection criteria, and eligibility, which is commendable. However, it is not clear why key hormonal regulators associated with metabolic syndrome, such as cortisol and the hypothalamic-pituitary-adrenal (HPA) axis, were not addressed or discussed. Considering the important role of cortisol in stress responses, inflammation, and metabolic regulation, it would be valuable to clarify whether relevant data were unavailable or excluded, or if this represents a potential area for future investigation.
Respond to the reviewer’s comment: Thank you for your valuable comment regarding the lack of crucial hormonal regulators, specifically cortisol and the hypothalamic-pituitary-adrenal (HPA) axis, in our systematic review and meta-analysis. We appreciate your emphasis on this important aspect of metabolic regulation. We have included a statement in the "2.3. Data collection" subsection of the Methods to clarify the scope of our data collection concerning hormonal parameters.
“The primary outcome measures for further analysis included direct metabolic parameters (e.g., body weight, glucose, insulin, lipid profiles), inflammatory markers, and oxidative stress status, which were based on the availability of data from the included studies. While cortisol levels and parameters related to the hypothalamic-pituitary-adrenal (HPA) axis are recognized as key factors in regulating metabolic syndrome, a comprehensive review of the included preclinical studies showed no findings regarding these hormonal outcomes. Consequently, cortisol and HPA axis parameters were excluded from this current review. It is important to emphasize that a comprehensive understanding of the role of kefir in modulating stress-related metabolic dysfunction is urgently needed.”
- Results
3.1. In the well-designed Figure 4, correct a minor typo in the third image where the word “genetic” is missing an “e.”. Additionally, I believe that it would be helpful to explicitly state that one model represents only diet-induced metabolic syndrome, whereas the other two involve diet as well as chemical or genetic induction methods. This clarification would enhance reader understanding of the experimental approaches compared.
Respond to the reviewer’s comment: The typing error in Figure 4 has been corrected, and the figure legend has been rewritten as follows to enhance clarity.
“Fig. 4. Types of rodents used in the included studies (A) and the detailed methods (B and C) for inducing metabolic syndrome in these rodents. The first group represents diet-induced metabolic syndrome, while the second and third models combine dietary and chemical induction (e.g., high-fat diet plus streptozotocin) and dietary and genetic induction (e.g., ApoE knockout mice), respectively. This classification reflects the experimental strategies utilized in the included studies.”
3.2. In Table 1, it would be helpful to include a dedicated column for references or to change some way, so that each study can be directly traced back to its source. Additionally, the purpose of the horizontal lines separating certain entries (e.g., between rows 8 and 9, 24 and 25, or 32 and 33) is unclear — if these lines indicate grouping or categorization, it would be useful to clarify this in a legend or footnote. Finally, the column “Types of treatment” is somewhat ambiguous, particularly the meaning of the terms in parentheses. I recommend clarifying this terminology or providing a short explanatory note to aid reader comprehension. Generally, it is a little bit confused.
Respond to the reviewer’s comment: The information and references have been added to Table 1 as suggested.
3.3. In Figure 6, the purpose of the upper panel titled “Treatment duration time” is not entirely clear. It might be helpful to explain why this information is presented separately, and how it contributes to the interpretation of the results. Additionally, in the figure legend, the phrase “stress status (A lower panel)” could be revised for clarity, as the reference to “A” is somewhat confusing — especially if there is no corresponding “B” panel. Furthermore, in the table summarizing inflammatory markers, IL-8 is included as a reported outcome, but it is not discussed anywhere in the main text. I recommend either referencing IL-8 explicitly in the results or discussion section, or removing it from the summary if not supported by the reviewed studies.
Respond to the reviewer’s comment: The figure has been edited, and the figure legend has been rewritten to enhance clarity. Furthermore, we rechecked the information regarding IL-8; there is no additional information extracted that reflects a change in IL-8. Therefore, the data was removed from this manuscript.
“Fig. 6. Overview of study characteristics and outcome measures. The upper panel summarizes the distribution of treatment durations across the included studies, while the lower panel presents the inflammatory and oxidative stress markers evaluated in those studies.”
3.4. In subsection 3.6, the sentences “A total of 38.58% (n = 12) of the studies indicated that various dietary patterns, including fructose, HFD, HFHF, and HFCS, significantly elevated levels of TNF-α. Whereas the intervention with kefir was found to reduce these levels” could be rephrased for improved clarity. I suggest merging the two parts into a single sentence to enhance reader comprehension.
Respond to the reviewer’s comment: The sentences were rewritten as follows.
“A total of 38.58% (n = 12) of the studies reported that dietary interventions such as fructose, HFD, HFHF, and HFCS led to significant increases in TNF-α levels, which were subsequently attenuated by kefir treatment.”
- Discussion
4.1. In the third paragraph, the sentence “in this review…pathways” ends with “which is possibly mediated by activating the PPARγ and AMPK pathways” should be revised. This conclusion is not clearly supported by the overall content of the review. Please clarify whether this statement is based solely on references 72 and 73. If so, this should be explicitly stated; otherwise, consider rephrasing or providing additional references to support this mechanistic claim.
Respond to the reviewer’s comment: The paragraph has been rewritten to improve clarity. Appropriate references were added to support this mechanistic claim.
“This review highlights studies that have demonstrated the influence of probiotics and peptides derived from kefir on weight gain, which may involve the modulation of metabolic pathways, such as PPARγ and AMPK. Additionally, kefir, kefir-derived probiotics, and peptides possibly exert these effects by promoting beneficial shifts in gut microbiota and enhancing the production of SCFAs, particularly butyrate [22,26,36,54]. While direct evidence from included animal studies remains limited, previous in vitro studies suggest that SCFAs, which are metabolites enhanced by probiotic fermentation, can activate peroxisome proliferator-activated receptor gamma (PPARγ) [72] and AMP-activated protein kinase (AMPK) pathways [73]. Kefir-derived bioactive peptides have been shown to activate AMPK signaling and upregulate PPARγ expression in cultured hepatocytes and adipocytes, contributing to enhanced lipid oxidation and decreased lipogenesis [24, 36, 37].”
4.2. In the fourth paragraph, in the sentence beginning with “Furthermore…conditions,” please include appropriate references in line with the referencing approach followed in the rest of the manuscript. Additionally, the abbreviation “HbA1c” should be written in full (i.e., glycated hemoglobin or hemoglobin A1c) upon its first mention for clarity and consistency.
Respond to the reviewer’s comment: Appropriate references were included as suggested. Furthermore, the abbreviation “HbA1c” was written in full and used consistently throughout the text.
4.3. In the sixth paragraph, the statement “Numerous studies have reported reduced levels of pro-inflammatory cytokines (TNF-α, IL-1β, and IL-6) in animals that were treated with kefir” is too vague. Please revise to specify which studies are being referred to, and include appropriate citations. Greater detail on the experimental context (e.g., animal models, dosages, or outcomes) would strengthen the scientific rigor of the manuscript.
Respond to the reviewer’s comment: The paragraph has been revised to enhance clarity as follows.
“Multiple preclinical studies demonstrate that kefir administration reduces pro-inflammatory cytokine levels, including TNF-α, IL-1β, and IL-6, in rodent models of metabolic syndrome. For example, Akar et al. (2021) [26] and Chang et al. (2023) [45] reported significant decreases in TNF-α and IL-1β levels in high-fructose-fed and atherogenic diet-fed rats and ApoE knockout mice, respectively, after kefir or kefir peptide supplementation. Kim et al. (2017) [25] and Kim et al. (2021) [37] observed mixed outcomes, with reductions in IL-6 but varying effects on TNF-α and IL-1β in high-fat diet-induced obese mice treated with kefir-derived probiotics. Similarly, Santanna et al. (2017) [62] and Tung et al. (2020) [49] found significant decreases in TNF-α and IL-6 levels following kefir intervention in LDL receptor-deficient mice and diet-induced obesity models. These studies involved various kefir formulations, including whole kefir, isolated peptides, and kefir-derived probiotics, administered over treatment periods ranging from 3 to 16 weeks at doses between 0.05 mg/g and 22 mL/kg body weight. “
Reviewer 3 Report
Comments and Suggestions for Authors
The systematic review titled “The impact of kefir consumption on inflammation, oxidative stress status, and metabolic syndrome-related parameters in animal models: A systematic review and meta-analysis” corresponds to a study of the systematic review and meta-analysis type in the area of traditional functional foods, specifically ancestral fermented foods like the dairy ferment Kefir. In this context, the manuscript is situated in the realm of probiotic foods and their evidence related to health effects, specifically their effects on health disorders, which in this case correspond to metabolic syndrome.
In this theoretical framework, the authors of the manuscript focus on generating an information analysis that addresses the existing gap in clinical studies in humans with scientific evidence of kefir consumption on effects in non-communicable chronic diseases. To this end, the manuscript appropriately centers on conducting a systematic review and meta-analysis of preclinical studies with rodents. In this scope, the manuscript reviews existing preclinical studies related to the use of kefir associated with the metabolic syndrome disorder.
Methodologically, the manuscript is a systematic review and uses appropriate tools such as PRISMA associated with CAMARADES and SYRCLE. In total, the methodology allows for a quantitative analysis of 38 original studies published, covering 1462 rodents, analyzing parameters: body weight, triglycerides, or insulin, associated with metabolic syndrome. The analysis covers the time period from 2011 to 2023.
The results and conclusions are coherent and appropriate to the information collected and analyzed, highlighting the heterogeneity of protocols without affecting the evidence related to the significance obtained in the positive evaluation of all parameters in animals exposed to kefir consumption.
It is suggested to improve the resolution of figures 3, 4, and 6 in the pie charts.
Author Response
The systematic review titled “The impact of kefir consumption on inflammation, oxidative stress status, and metabolic syndrome-related parameters in animal models: A systematic review and meta-analysis” corresponds to a study of the systematic review and meta-analysis type in the area of traditional functional foods, specifically ancestral fermented foods like the dairy ferment Kefir. In this context, the manuscript is situated in the realm of probiotic foods and their evidence related to health effects, specifically their effects on health disorders, which in this case correspond to metabolic syndrome.
In this theoretical framework, the authors of the manuscript focus on generating an information analysis that addresses the existing gap in clinical studies in humans with scientific evidence of kefir consumption on effects in non-communicable chronic diseases. To this end, the manuscript appropriately centers on conducting a systematic review and meta-analysis of preclinical studies with rodents. In this scope, the manuscript reviews existing preclinical studies related to the use of kefir associated with the metabolic syndrome disorder.
Methodologically, the manuscript is a systematic review and uses appropriate tools such as PRISMA associated with CAMARADES and SYRCLE. In total, the methodology allows for a quantitative analysis of 38 original studies published, covering 1462 rodents, analyzing parameters: body weight, triglycerides, or insulin, associated with metabolic syndrome. The analysis covers the time period from 2011 to 2023.
The results and conclusions are coherent and appropriate to the information collected and analyzed, highlighting the heterogeneity of protocols without affecting the evidence related to the significance obtained in the positive evaluation of all parameters in animals exposed to kefir consumption.
It is suggested to improve the resolution of figures 3, 4, and 6 in the pie charts.
Respond to the reviewer’s comment: The resolution of the mentioned pie charts has been improved as suggested.